# Hereditary Thrombotic Thrombocytopenic Purpura

**DOI:** 10.3390/genes14101956

**Published:** 2023-10-18

**Authors:** Sanober Nusrat, Kisha Beg, Osman Khan, Arpan Sinha, James George

**Affiliations:** 1Hematology-Oncology Section, Department of Internal Medicine, University of Oklahoma Health Sciences Center, Oklahoma City, OK 73104, USA; 2Jimmy Everest Section of Pediatric Hematology-Oncology, Department of Pediatrics, University of Oklahoma Health Sciences Center, Oklahoma City, OK 73104, USA

**Keywords:** hereditary thrombotic thrombocytopenic purpura, thrombotic thrombocytopenic purpura, thrombotic microangiopathies, recombinant ADAMTS13, rADAMTS13

## Abstract

Hereditary thrombotic thrombocytopenic purpura (hTTP), also known as Upshaw–Schulman syndrome, is a rare genetic disorder caused by mutations in the ADAMTS13 gene that leads to decreased or absent production of the plasma von Willebrand factor (VWF)-cleaving metalloprotease ADAMTS13. The result is circulating ultra-large multimers of VWF that can cause microthrombi, intravascular occlusion and organ damage, especially at times of turbulent circulation. Patients with hTTP may have many overt or clinically silent manifestations, and a high index of suspicion is required for diagnosis. For the treatment of hTTP, the goal is simply replacement of ADAMTS13. The primary treatment is prophylaxis with plasma infusions or plasma-derived factor VIII products, providing sufficient ADAMTS13 to prevent acute episodes. When acute episodes occur, prophylaxis is intensified. Recombinant ADAMTS13, which is near to approval, will immediately be the most effective and also the most convenient treatment. In this review, we discuss the possible clinical manifestations of this rare disease and the relevant differential diagnoses in different age groups. An extensive discussion on prophylaxis and treatment strategies is also presented. Unique real patient cases have been added to highlight critical aspects of hTTP manifestations, diagnosis and treatment.

## 1. Introduction

Hereditary thrombotic thrombocytopenic purpura (hTTP), also known as Upshaw–Schulman syndrome, is a rare genetic disorder with possible lifetime morbidity and early mortality [1]. The estimated prevalence is one case per million people. It is an autosomal recessive disorder caused by mutations in the *ADAMTS13* gene that leads to decreased or absent production of the plasma von Willebrand factor (VWF)-cleaving metalloprotease ADAMTS13. The result is circulating ultra-large multimers of VWF that can cause microthrombi, intravascular occlusion and organ damage, especially at times of turbulent circulation. Patients with hTTP may have many overt or clinically silent manifestations. A high index of suspicion is required for diagnosis. Times of great risk are newborn infancy and pregnancy. Recognition of hTTP and institution of early prophylactic treatment can reduce long-term complications. The aim of this review is to describe the pathophysiology, symptomatology in different vulnerable age groups and the available treatment options to provide physicians with the means to suspect hTTP and refer to hematologists for appropriate diagnosis and management.

Because hTTP is rare, it is often initially misdiagnosed as the much more commonly acquired autoimmune TTP (iTTP) [2]. The incidence of iTTP is 2–3 cases per million people in a year. Diagnosis of iTTP is confirmed by identification of anti-ADAMTS13 antibodies, but these antibodies are not always detectable.

## 2. Pathophysiology

hTTP is an autosomal recessive condition characterized by presence of biallelic pathogenic variants in the *ADAMTS13* (a disintegrin and metalloproteinase with a thrombospondin type 1 motif, member 13) gene located on chromosome 9q34 [3,4]. The normal gene codes for ADAMTS13, which is an active enzyme responsible for the proteolysis of the ultra-large von Willebrand factor (UL-VWF) multimers [5,6]. When endothelium is activated, the severe deficiency of ADAMTS13 in hTTP results in the accumulation of these large multimers with an increased ability to bind platelets. Platelet-attached VWF multimers cause a further increase in shear stress in the microvasculature [7]. An abnormal coagulation cascade, microvascular thrombi and hemolysis thus ensue [8] (Figure 1).

Over 200 pathogenic variants in the *ADAMTS13* gene have been demonstrated in various studies [9,10,11]. In the International Hereditary Thrombotic Thrombocytopenic Purpura Registry, c.4143_4144dupA (exon 29;p.Glu1382Argfs*6) was the most frequent mutation present, present on 60 of 246 alleles [9]. In 2019, van Dorland et al. evaluated the residual ADAMTS13 activity and its relationship to disease onset with an emphasis on carriers with c.4143_4144dupA mutation. They reported that homozygous status or residual ADAMTS13 activity did not appear to have a significant impact on disease phenotype. In fact, a greater number of compound heterozygotes developed overt disease manifestations at an earlier age while having higher baseline activity of ADAMTS13 [9]. Conversely, in a cohort of 29 hTTP patients, Lotta et al. found that residual ADAMTS13 activity < 3% was associated with earlier disease onset [12]. They also described a subset of patients with homozygous R1060W genotype who had a higher residual ADAMTS13 activity and later age at onset of first TTP episode requiring therapeutic intervention. This depicts the inconsistencies seen in genotype–phenotype correlation in hTTP. Together with the influence of other intrinsic factors, severity of disease can vary widely between individuals regardless of the underlying genetic mutations present.

## 3. Clinical Presentation and Differential Diagnoses

### 3.1. Newborn Infants

hTTP often presents with severe hyperbilirubinemia, caused by hemolysis, and thrombocytopenia within the first days after birth. The diagnosis of hTTP is often not considered due to its rarity. It is often misdiagnosed as the more common ABO incompatibility, which can cause milder hyperbilirubinemia [6,13,14]. The first hours of life are critical for infants with hTTP because the turbulent circulation in the patent ductus arteriosus causes uncoiling of the UL-VWF multimers, allowing for platelet binding and subsequent platelet aggregation [15], causing systemic microvascular thrombosis. There are no fetal complications before birth because there is no turbulent circulation in the fetus. Liu et al. were the first to present quantitative data to distinguish the severe hemolysis and hyperbilirubinemia of hTTP from ABO incompatibility (Table 1) [16,17]. In this retrospective comparative case series, all four patients with hTTP developed jaundice within 24 h and the maximum serum bilirubin concentrations were higher and developed earlier compared to neonates with ABO incompatibility (median, 24 mg/dL at 38 h after birth versus 16 mg/dL at 74 h after birth). Additionally, anemia and severe thrombocytopenia were common in hTTP patients, all hTTP patients having severe thrombocytopenia. Serum creatinine and blood urea nitrogen were also higher in the hTTP patients. Not surprisingly, all four neonates with hTTP demonstrated refractoriness to phototherapy and intravenous immunoglobulin. However, they responded to whole blood exchange transfusion. Two of the four infants had older siblings who died at birth, undoubtedly with unrecognized hTTP. None of the four infants were diagnosed with hTTP until they later experienced recurrent symptoms at 1–48 months old.

In a review of publications describing 208 patients with hTTP [18], 9 of 32 deaths occurred in neonates. Only one infant was suspected to have hTTP before death, but he was not treated; two were suspected and confirmed to have hTTP after death; six were only suspected after a subsequent sibling was born and diagnosed with hTTP. This illustrates that hTTP is rarely recognized at birth and highlights the importance of suspecting hTTP as one of the potential etiologies of neonatal hemolysis and jaundice, especially when concomitant thrombocytopenia is present.

Because it is rare, hTTP is often not considered as a potential cause for neonatal hyperbilirubinemia, even if is accompanied by thrombocytopenia. Confounders like critical illness, preeclampsia-causing thrombocytopenia, and anemia are common especially in newborn infants admitted to the neonatal intensive care unit. This leads to delayed or missed diagnosis of hTTP. This must change, and a high index of suspicion for hTTP can certainly lead to improved outcomes. If hTTP is suspected, the current treatment regimen is plasma infusion or a factor VIII concentrate that contains ADAMTS13 (e.g., Koate, Octaplas^®^). When recombinant ADAMTS13 (rADAMTS13) becomes available, it will allow for prompt, complete recovery [19]. Subsequent confirmation of the diagnosis of hTTP is established by documenting ADAMTS13 deficiency. Florescence resonance energy transfer (FRET) assay is the most commonly available commercial method for quantification of ADAMTS13 activity. The cases described in this review had testing completed using this assay.

Other hemolytic disorders in newborn infants are characterized by the presence of progressively worsening hyperbilirubinemia and anemia within the first few weeks of life [20]. The most common cause of neonatal hyperbilirubinemia is ABO incompatibility, in which maternal antibodies cross the placenta to attack fetal red blood cells. In contrast to hTTP, the hyperbilirubinemia is not severe, and it resolves with phototherapy [21]. In a study of 878 deliveries, 17.3% neonates were at risk for hemolysis because they had blood groups A or B and were born to mothers with blood group O [22]. One-third of these neonates developed jaundice requiring phototherapy. Neonates who have red cell antigen incompatibility can be identified with a positive direct antiglobulin test [21].

**Case 1: A Case of hTTP in a Newborn Infant [19]** 

A pregnant woman had four previous pregnancies: two had been uncomplicated; one infant died 3 days after delivery; a subsequent infant died 14 days after delivery. Both of these infants had severe hyperbilirubinemia and thrombocytopenia. After the second infant’s death, pathogenic biallelic *ADAMTS13* mutations were documented. In this report of a subsequent delivery [19], the infant was normal at birth, but her platelet count decreased to 13,000/µL in 7 h. She was managed with Octaplas^®^ (solvent detergent fresh frozen plasma) that maintained her platelet count at 30,000/µL until rADAMTS13 was available (on a compassionate use basis) at 24 h. After 3 days of rADAMTS13 treatment, her platelet count was normal. Subsequently, the diagnosis of hTTP was confirmed by ADAMTS13 activity <2%. After day 6, rADAMTS13 was continued every 10 days. At age 2, she had reached all developmental milestones and had had no symptoms of TTP. This experience emphasizes the role of early diagnosis and treatment in the management of hTTP to prevent neonatal morbidity and mortality. If hyperbilirubinemia is severe, one should consider hTTP as a possibility in the list of potential diagnosis. In the future, availability of rADAMTS13 will provide prompt, durable recovery.

### 3.2. Children

hTTP in children may have a subtle presentation. Children may remain asymptomatic for years, or they may have minor symptoms (e.g., headache, lethargy) and occasional acute exacerbations [23]. Isolated thrombocytopenia is also a common manifestation of hTTP in childhood and is frequently mistaken for immune thrombocytopenic purpura [24]. Evaluation of thrombocytopenia in children should always include examination of the peripheral blood smear to look for schistocytes, which suggest occurrence of microangiopathic hemolytic anemia (MAHA). The presence of schistocytes or frequent recurrences of thrombocytopenia should prompt providers to obtain ADAMTS13 levels. Stroke, transient ischemic attacks (TIAs) or transient neurologic symptoms may occur in children with hTTP [9,24]. In a systematic review of case reports on infants and children with hTTP, 15 of 69 individuals experienced either a TIA or a stroke [23]. The age of occurrence of strokes in children with hTTP is similar to the age of occurrence of strokes in patients with sickle cell disease [25]. Furthermore, long-term consequences from stroke in the form of cognitive impairment, memory, depression and anxiety are common [26] and in some cases these occur without overt stroke [27].

Before puberty, iTTP is less common than hTTP and the frequency of iTTP among girls and boys is similar. After ages 9 and older, the incidence of iTTP increases dramatically, and girls are more commonly affected, with an 81% female preponderance. (Figure 2) [23]. This is consistent with the Oklahoma iTTP Registry data: Out of 90 patients included, only 2 of 90 patients are children (<18 yo). Seventy-five percent of patients with iTTP are women. Also, 35 (39%) of these 90 patients are African American [28]. The predominance of girls among children with iTTP following puberty is similar to systemic lupus erythematosus (SLE). The clinical features of TTP are also similar to the clinical features of SLE [29]. The increased frequency in patients of African descent with iTTP is also similar to SLE. SLE and iTTP are both autoimmune disorders and both occur predominantly in young, Black women. Interestingly, the occurrence of SLE among patients with iTTP is also common [30,31].

**Case 2: A Case of Delayed Diagnosis of iTTP in a Child** 

A 9-year-old boy presented with 2 days of intermittent abdominal pain and vomiting. The physical examination, including neurological examination, was normal. His platelet count was 12,000/microliter, hematocrit was 10%, serum creatinine was 0.7 mg/dL, serum lactate dehydrogenase (LDH) was 3433 U/L, and the peripheral blood smear demonstrated schistocytes. He was diagnosed with typical HUS despite the absence of diarrhea and kidney failure, because in children, this is the most common cause of microangiopathic hemolytic anemia and thrombocytopenia. He was managed only with transfusions. Over the next 14 days, he received 14 units of red cells to maintain his hematocrit at 17–25% and 13 platelet transfusions to maintain his platelet count at 7000–14,000/µL. His serum creatinine remained less than 1.0 mg/dL. During these 2 weeks, he had several episodes of transient diplopia. The diagnosis of TTP was finally considered and plasma exchange was started 14 days after admission. ADAMTS13 activity was 7% with an absent inhibitor. hTTP was considered because he was a child and there was no detectable ADAMTS13 inhibitor. However, the absence of an inhibitor may have been caused by his many previous transfusions. The diagnosis of iTTP was established when he relapsed one year later and his ADAMTS13 activity was <5% an inhibitor of 0.9 BU. He was treated successfully with plasma exchange and corticosteroids. His ADAMTS13 activity following recovery was >100%.

### 3.3. Adults

While the occurrence of thrombocytopenia, MAHA and neurologic symptoms may begin early in childhood, clinical diagnosis is often not made until adulthood [32]. In adults, excessive alcohol consumption may be a trigger for acute episodes of TTP, mostly occurring in men [9]. Other triggers include inflammatory conditions, trauma and drugs. Usual manifestations are findings of MAHA and thrombocytopenia. In addition, neuropsychiatric symptoms can be present; these include headaches, poor concentration and depression in over 60% of patients and lead to a substantial disease burden [27]. Myocardial infarction is uncommon. With increasing age, kidney failure can occur. Ten percent of patients from the international hTTP registry required renal replacement therapy and 2% underwent renal transplant [9].

Among young adults, the diagnosis of hTTP is principally in women during pregnancy (Figure 3) [18]. Forty-two percent of United Kingdom (UK) hTTP registry patients had their initial clinical exacerbation during pregnancy [24]. A review of the frequency and severity of hTTP exacerbations in women during pregnancy, which described 61 pregnancies in 35 women, demonstrates that exacerbations during pregnancy may be inevitable [33]. The clinical data were valid because the objective of these case reports was to describe novel *ADAMTS13* mutations. Thirty-four (97%) women had severe complications of pregnancy, defined as requiring urgent hospitalization. Two of these women died as a consequence of TTP complications.

The complications during pregnancy are assumed to be caused by turbulent blood circulation in the placenta. In uncomplicated pregnancies of healthy women, the maternal spiral arterioles become dilated, allowing for increased blood flow with slower velocity. The failure of this transformation leads to maternal vascular malperfusion, which is the etiology of preeclampsia [34]. The placental pathology of severe preeclampsia is decidual arteriolar thrombosis with placental infarction. Placental pathology of women with hTTP is the same as the placental pathology of severe preeclampsia [35]. The inevitable complications of pregnancy are the cause of the increased frequency of hTTP among women (Figure 3).

hTTP can be misdiagnosed as preeclampsia or HELLP syndrome. The diagnosis of hTTP should be considered in pregnant women who develop severe preeclampsia or HELLP syndrome. Only 6% of preeclampsia occurs before 30 weeks’ gestation, and only 6% of women with preeclampsia have platelet counts <30,000/µL [36]. Therefore, it is critically important to consider hTTP and measure ADAMTS13 activity in these women.


**Case 3: Pregnancy in a Woman with hTTP [37]**


A 27-year-old woman was first seen at 31 weeks’ gestation in her second pregnancy. Twelve months earlier, her first pregnancy had ended in the 22nd gestational week with an intrauterine fetal death, which had been preceded by progressive maternal thrombocytopenia and intrauterine growth restriction. Two weeks after the fetal death, the patient had an ischemic stroke. Her platelet count was 40,000/µL. Antiphospholipid syndrome was suspected, but repeated testing for antiphospholipid antibodies was negative. A diagnosis of seronegative antiphospholipid syndrome was made. When she became pregnant for the second time, prophylactic treatment with dalteparin was started. The pregnancy was uneventful until gestational week 30, when she had a second stroke. The dalteparin dose was increased and treatment with hydroxychloroquine was started. The patient was in overall good health but had residual facial palsy and slurred speech. Blood pressure, heart rate and temperature were normal, and she had no edema. An ultrasound evaluation showed a viable but small-for-gestational-age fetus. Her platelet count was 92,000/µL; serum LDH was mildly increased. Hemoglobin, serum haptoglobin, serum creatinine and liver enzymes were normal. ADAMTS13 activity was 3.9%, confirming the diagnosis of TTP. Treatment for iTTP with plasma exchange and corticosteroids was started. Dalteparin was reduced to the prophylactic dose, and hydroxychloroquine was stopped. Subsequently, hTTP was diagnosed when no ADAMTS13 inhibitor was identified. The patient’s parents described that the patient had undergone a neonatal whole blood exchange transfusion because of severe hyperbilirubinemia soon after birth. Despite plasma exchange, her platelet count remained <70,000/µL. Then, rADAMTS13 (on a compassionate use basis) was started weekly at 33 weeks’ gestation. Her platelet count increased to 190,000/µL. She delivered a healthy baby boy at 37 weeks’ gestation by Cesarean section, and then continued treatment with rADAMTS13. Placental histology documented maternal vascular malperfusion.

**Case 4: Different Clinical Features in Siblings with hTTP** 

A 37-year-old woman had exhibited no symptoms at the time of her birth but had symptoms of depression during elementary school. At age 16, she had thrombocytopenia that was diagnosed as ITP; at age 17, she had a TIA; at 21, she had a stroke. She was pregnant at age 24 and was diagnosed with severe preeclampsia, requiring emergency Cesarean section delivery of a healthy infant. She had recurrent strokes at ages 28, 33, 34 and 37. hTTP was finally diagnosed at age 37 after ADAMTS13 activity was found to be <5% and ADAMTS13 sequencing confirmed the diagnosis. She was started on regular biweekly plasma infusions, and she was enrolled into the international hTTP registry. ADAMTS13 sequencing was also performed for her parents and brother. He has the same pathogenic biallelic mutations as his sister, ADAMTS13 activity < 5%; his medical evaluation, including brain MRI, was normal. Her brother, age 33, has had no health issues. Siblings with hTTP often have different clinical features, but this occurrence of very severe symptoms in one sibling and no symptoms in the other is rare. One of this family’s pathogenic mutations, R1060W, is common and is associated with residual ADAMTS13 activity. This story illustrates that the pathogenesis of hTTP is complex, involving much more than ADAMTS13 deficiency. hTTP is different from hereditary disorders such as hemophilia, in which the clinical features are predictable from the factor VIII level.

## 4. iTTP

iTTP is a disorder in young adults; the median age of diagnosis is 40 [28]. Although iTTP is often described as a disorder of acute episodes and remissions, remission may not mean recovery. As with other autoimmune disorders, iTTP is a lifetime disorder. Even if acute episodes do not recur, asymptomatic immune-mediated ADAMTS13 deficiency can be dangerous. Some patients may recover normal ADAMTS13 activity, with apparently stable values of 100%. But many patients with iTTP who are continuously asymptomatic may have lower than normal ADAMTS13 activity. These patients, like the heterozygous siblings and parents of patients with hTTP, are likely at risk for stroke and other cardiovascular disorders. This risk was predicted by the Rotterdam study of normal subjects. Subjects in the lowest quartile of normal ADAMTS13 activity had twice the risk for stroke (7.3%) as subjects in the highest quartile (3.6%) [38]. Increased risk for stroke in patients with iTTP in remission but with low ADAMTS13 activity was recently confirmed. Among 36 patients with iTTP in clinical remission, patients with lower ADAMTS13 activity had higher risk for stroke [39].

## 5. Diagnostic Evaluation

When TTP is suspected, emergent laboratory testing should include complete blood count (CBC), reticulocyte panel, serum haptoglobin level, serum lactate dehydrogenase level, direct antiglobulin test, indirect bilirubin, serum creatinine, blood smear evaluation for schistocytes, coagulation studies including prothrombin time, activated partial thromboplastin time, serum fibrinogen level, and an ADAMTS13 activity and inhibitor level. Both hTTP and iTTP are characterized by the presence of severe thrombocytopenia and evidence of microangiopathic hemolytic anemia (hemolysis and schistocytes on blood smear). An ADAMTS13 level less than 10% with an absent inhibitor, or a positive family history of TTP, may guide the clinician towards an inherited rather than an immune etiology of TTP. The definitive diagnosis requires genetic testing to show the presence of biallelic pathogenic mutations in the *ADAMTS13* gene.

## 6. Management Strategies

The goals of treatment for hTTP and iTTP are distinct. For hTTP, the goal is simply replacement of ADAMTS13. The primary treatment is prophylaxis, providing sufficient ADAMTS13 to prevent acute episodes. When acute episodes occur or risk is great, as with pregnancy, the prophylaxis is intensified. For iTTP, the goal is suppression of anti-ADAMTS13 antibodies or removal by plasma exchange. Providing ADAMTS13 by replacement or preventing thrombosis with caplacizumab are secondary treatments to allow time for immunosuppression to succeed.

When children or adults with hTTP have acute symptoms, such as TIA or stroke, plasma infusion is appropriate and effective therapy. The more important issue for patients with hTTP is how to determine the need for prophylactic treatment to prevent TIA or stroke. Some patients with hTTP seem to be normal for many years, with no symptoms. Women with hTTP may be normal except during a pregnancy. Recent data show that patients with iTTP in remission might experience cognitive symptoms and are also susceptible to TIA and stroke. These findings imply that severe deficiency of ADAMTS13 is a risk for cognitive impairment [39]. This suggests that prophylactic treatment may be appropriate for all patients with hTTP.

Current prophylactic treatment strategies are using plasma and plasma-derived factor VIII concentrates that contain sufficient ADAMTS13 (Koate, Octaplas^®^). Plasma prophylaxis can be burdensome to some patients, as it requires them to dedicate several hours of their time to therapeutic infusions at a specialized center every two weeks. Some patients may even need weekly plasma infusions to prevent recurrent symptoms. The factor VIII concentrates can be self-administered at home, which is a considerable benefit. However, the amount of ADAMTS13 in the factor VIII concentrates may be small, and frequent infusions, more often than weekly, may be needed. rADAMTS13, which is near to approval, will immediately become the most effective and also the most convenient treatment. An early-phase clinical trial, the outcomes of which are accessible [40], showed that that ADAMTS13 levels of 100% can be achieved with infusion of <10 mL, and infusions at 2–3-week intervals may be sufficient.

When rADAMTS13 is available, prophylactic treatment may expand to most patients. Asymptomatic patients may not need prophylactic rADAMTS13, but very careful medical follow-up is essential. Symptoms of hTTP may be subtle. They may be mood issues, depression, headaches or other symptoms that do not seem dangerous. As asymptomatic patients get older, some subtle symptoms may be inevitable. And, of course, the first symptom could be a stroke. The difficult question for treatment of hTTP may be whether rADAMTS13 prophylaxis should begin at the time of diagnosis.

There are three main categories of therapeutic agents for hTTP, which include plasma infusions, plasma-derived factor VIII concentrates and newer treatments such as recombinant ADAMTS13. The use of caplacizumab in an isolated case of refractory hTTP and groundwork data on novel gene therapy is also described here (Figure 4).

### 6.1. Plasma

Plasma infusions are the mainstay hTTP treatment and are used both in acute episodes and for prophylaxis. They replenish ADAMTS13, providing cleavage of the UL-VWF multimers. Patients with hTTP respond promptly to plasma infusion. Infusion of plasma carries associated risks including immunogenic reactions that can be severe and increases health care utilization due to need for specialized infusion center services. Administration requires venipuncture or placement of an infusion port, with the latter increasing the risk of thrombosis and blood stream infections. Patients need plasma infusions anywhere between 7 and 21 days and the dose is 10–15 mL/kg. Alwan et al. reported from their hTTP registry that the 3-weekly regimen of plasma infusions was insufficient in 70% of patients and weekly/fortnightly infusions were required to see the benefit [24]. The half-life of ADAMTS13 for patients receiving regular plasma infusions is variable; reports range between 2 and 5 days. The half-life is dependent on weight, metabolism and the severity of the disease. A report on the pharmacokinetics of plasma infusions describes that patients returned to their baseline ADAMTS13 level within 7–10 days post plasma infusion [42]. The ADAMTS13 activity needed to prevent attacks is unknown; however, aiming to correct to normal may not be feasible, except with recombinant ADAMTS13. The case series presented by Taylor et al. concluded that targeting an ADAMTS13 trough activity of 10 IU/dL is more feasible than targeting for a higher ADAMTS13 activity of 50 IU/dL due to the increased volume and frequency of infusions required with the latter [42].

Patients with hTTP respond quickly to treatment during acute episodes; however, the role of long-term prophylaxis is not clear. Given this lack of information there is some hesitancy to treat patients with hTTP in remission with prophylactic therapy. Tarasco et al.’s prospective cohort brought into question the utility of prophylactic treatment. The annual reported incidence of acute episodes of hTTP was similar for those on prophylaxis, 0.36, versus 0.41 for patients who did not receive prophylaxis [32]. This apparent failure of plasma prophylaxis may be related to insufficient volumes of plasma or insufficient frequency. However, given the risk of microvascular injury, especially in children, causing cognitive decline and eventual chronic kidney disease, there have been recommendations to start hTTP patients on prophylaxis. A report from a large cohort of hTTP patients from the United Kingdom demonstrated that prophylaxis not only abated subtle symptoms of microvascular injury (headache, abdominal pain, lethargy in patients with normal platelet count) but also significantly reduced the risk of stroke (2% vs. 17%, *p*-value = 0.04) in patients receiving prophylactic therapy [24]. After a detailed review of the literature, treatment recommendations were made by the International Society of Hemostasis and Thrombosis as a guideline document to help inform clinicians who manage these patients. These consensus guidelines recommend the use of prophylactic treatment with plasma versus a wait and watch strategy in hTTP patients in remission [43].

### 6.2. Plasma-Derived FACTOR VIII Concentrates

The use of plasma derived factor VIII concentrates has been reported in the past decade. One such report was made by Naik and colleagues about a 19-year-old who had been on FFP infusions for 15 years and then developed a severe reaction requiring admission to the intensive care unit and systemic desensitization without any improvement. The patient was treated with Koate and promptly achieved a platelet count > 100,000 and has continued to be in remission for 36 months [44].

There are two plasma-derived factor VIII concentrates that have been reported in the use of hTTP. One is Koate and the other includes BPL8Y [45]. Two independent laboratories conducted analysis of the content of ADAMTS13 in several factor VIII concentrates. They found the concentration of ADAMTS13 to be highest in Koate at 9.08 + 0.70 units/mL [46]. The reported dose in the literature for Koate is 30–50 IU/kg. Each 100 IU/mL of Koate provides 3–6 IU/kg of ADAMTS13. Reported frequency of infusion is from two times/week to once every 2–3 weeks [47]. A 5-year follow up of 11 patients, 9 of whom were receiving prophylactic infusions, did not report any long-term side effects, intolerance or difficulty with administration of Koate [48].

The benefits of Koate include lower volume of administration and lower cost due to self/caregiver administration. However, there have been concerns about long-term use in children due to limited long-term data. The ISTH 2020 guidelines recommend against the use of factor VIII concentrates due to variable concentrations of ADAMTS13; however, there has been growing evidence for its use since then [43].

### 6.3. Recombinant ADAMTS13

The newest treatment is recombinant ADAMTS13 (rADAMTS13), a product that has recently completed a phase 3 clinical trial. A large trial was conducted evaluating the safety and efficacy of prophylactic administration of rADAMTS13 in hereditary TTP. The interim analysis of this ongoing study showed promising results. All patients aged 0–70 were eligible for enrollment and were treated with 40 IU/kg of intravenous rADAMTS13 or standard of care (usually fresh frozen plasma at a dose of 10 mL/kg) in a crossover design administered every week or every other week. Efficacy from 38 adult and adolescent patients showed that no events occurred during treatment period with rADAMTS13, and one event occurred during treatment with standard care. All of the hTTP manifestations, including thrombocytopenia, renal dysfunction, neurological symptoms and abdominal pain, were evaluated. Of patients receiving rADAMTS13, 10.3% experienced treatment-related side effects versus 50% receiving standard care treatment. There were no therapy interruptions or discontinuation in the rADAMTS13 arm. However, 18.2% of patients in the standard of care arm had interruptions/discontinuation related to therapy. The time for which ADAMTS13 activity remained above 10% was 5.3 days in the rADAMTS13 arm vs. 1.7 days in the standard care arm. The variability of delivery of ADAMTS13 after treatment with rADAMTS13 was significantly lower compared with plasma infusions. There were no treatment-neutralizing or treatment-boosted antibodies reported during the study duration [49].

### 6.4. Immune-Modulating Therapies

Caplacizumab was developed as a treatment modality for acquired or immune TTP where anti-ADAMTS13 antibodies are the culprit. Caplacizumab is a nanobody that binds the A1 domain on VWF protein. This is the usual site where activated platelets bind via GP1b interaction. Once the binding site is blocked by caplacizumab, further interaction between ultra-large VWF proteins and platelets is inhibited regardless of the ADAMTS13 activity in vivo. A recent report described a 40-year-old male with hTTP who experienced multiple relapses despite every two-week plasma infusion and developed complications of fluid overload in the setting of chronic kidney disease. The patient did not respond to plasma infusions for one such relapse and declined plasma exchange. Caplacizumab was administered and plasma infusions were continued. Patient started to respond within 12 h and his renal function and platelets were back to baseline upon two-week follow up [50]. Use of caplacizumab in hTTP may be a consideration going forward. Even though no pediatric patients were enrolled on the caplacizumab trial, model-based dosing has been published recommending a daily dose of 5 mg in patients <40 kg and 10 mg in children weighing ≥40 kg [51]. Additionally, use of caplacizumab has been described in pediatric patients in some of the real-world experience literature, providing some tolerability data [52]. With the advent of recombinant ADAMTS13 therapy, caplacizumab has a limited role in hTTP. The authors of this review would recommend exploration of other strategies before using caplacizumab for hTTP.

### 6.5. Novel Therapies

There are some newer therapies in development. There are reports of safety and efficacy of gene transfer via adenovirus [53], lentivirus [54], and most recently via adeno-associated virus in ADAMTS-13 knock-out mice. Jin and colleagues described an animal model of ADAMTS13 ^−/−^ mice that were injected with an adenovirus-associated vector encoding for a murine ADAMTS13 variant with a liver specific promoter [55]. Two weeks later, the mice were injected with a shiga toxin strain known to induce a “TTP-like” syndrome. The ratio of high to low molecular weight VWF multimers in plasma was reduced in the mice treated with the vector, demonstrating that the vector was sufficient to reduce circulating UL-VWF multimers, which are present in ADAMTS13 ^−/−^ mice. The mice treated with the vector did not demonstrate severe thrombocytopenia (40–60% drop), which was noted in the knockout mice about 24–48 h after delivery of the shiga toxin. This adeno virus-associated type 8 vector has demonstrated safety in other genetic diseases, including hemophilia B [56] and Leber’s [57] congenital amaurosis. These observations suggest a future where gene therapy may be superior to plasma infusions in providing long-term remission.

### 6.6. Aspirin

Prevention of stroke and other thrombotic complications depend on the level of ADAMTS13 in the body and the level that aids in prevention of these complications has not been established. The need for recurring treatments with either plasma or rADAMTS13 makes it difficult to achieve a consistent trough level of ADAMTS13. That being said, other preventative strategies have been explored and postulated. Shao et al. described a mouse model of TTP that describes the mechanisms of clotting in TTP [58]. The mechanism of thrombosis in TTP requires not only the interaction of UL-VWF and platelets via the interaction of platelet receptor GP1bα with the A1 domain of VWF but also the activation of platelet fibrinogen receptor, αIIbβ3, thus allowing for fibrinogen binding and subsequent platelet aggregation resulting in microthrombi and thrombus propagation. The latter requires intracellular signaling via platelet C-type lectin-like receptor 2 (CLEC-2). Deletion of CLEC-2 inhibits the αIIbβ3 activation induced by VWF binding to platelets. The mice with CLEC-2 deletion or those treated with an αIIbβ3 antagonist had decreased pulmonary artery thrombosis and decreased severity of thrombocytopenia. Aspirin also prevents platelet aggregation by blocking the activation of platelet fibrinogen receptor αIIbβ3. These correlations suggest that aspirin may be effective in preventing thrombosis via inhibition of platelet aggregation [41].

## 7. Conclusions

Despite significant advancements in the genetics and management of hTTP over the last two decades, prompt diagnosis is still challenging due to clinical heterogeneity, including a substantial portion of patients presenting with subclinical manifestations and other confounding factors. A high index of suspicion and implementation of prompt treatment will continue to improve short- and long-term outcomes. Long-term data are needed for the newer therapies, including rADAMTS13. Additionally, although we are far away from gene therapy, it may offer a long-term cure for patients with hTTP.

## Figures and Tables

**Figure 1 genes-14-01956-f001:**
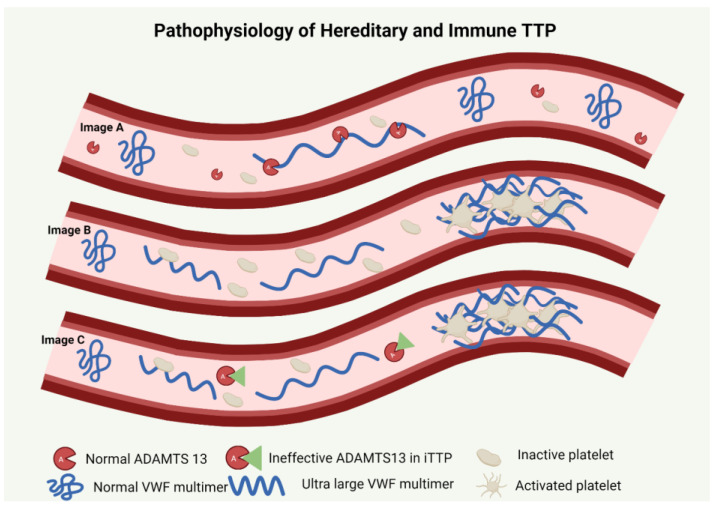
Pathophysiology of TTP. Image A depicts normal physiology with ADAMTS13-mediated cleavage of ultra-large VWF multimers. Image B describes pathophysiology of hTTP with absence of ADAMTS13 leading to circulating ultra-large VWF with sheer stress causing interaction of activated platelets and VWF multimers thus propagating microthrombi causing occlusion. Image C describes immune TTP where an anti-ADAMTS13 antibody renders the protein nonfunctional or enhances its clearance leading to circulating ultra-large VWF interacting with activated platelets causing microvascular thrombosis.

**Figure 2 genes-14-01956-f002:**
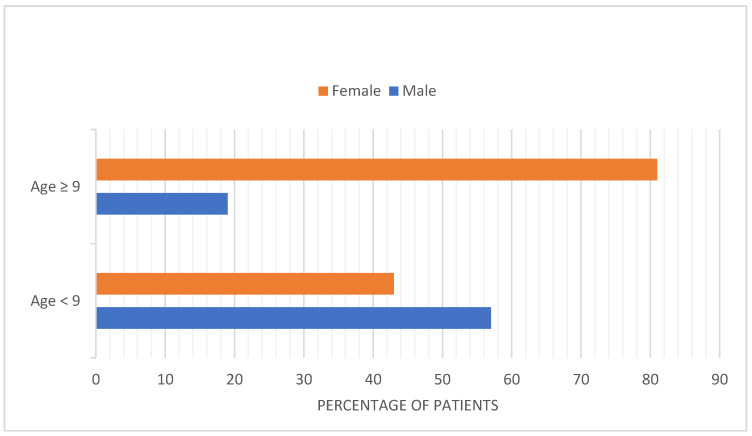
Frequency of diagnosis of immune thrombotic thrombocytopenic purpura (iTTP) amongst males and females before and after puberty [23]. Data used from publication by Siddiqui et al., *Pediatr Blood Cancer*, 2021.

**Figure 3 genes-14-01956-f003:**
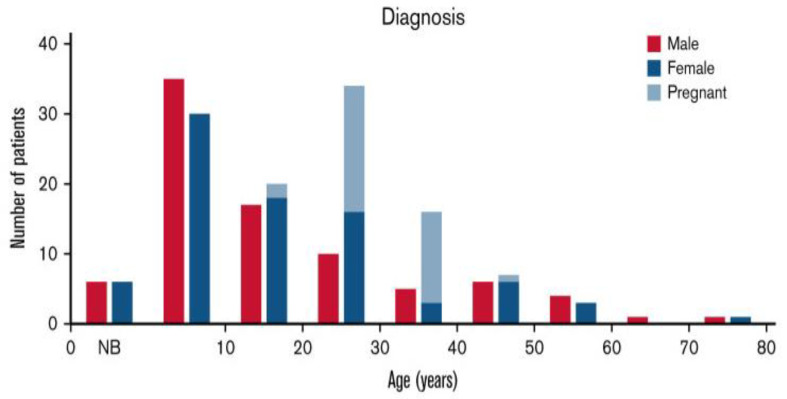
Age at diagnosis in patients with hereditary thrombotic thrombocytopenic purpura (hTTP). Age at diagnosis was reported for 202 (89%) of the 226 patients [18]. Figure reused from publication by Borogovac et al., *Blood Adv*, 2022 [18].

**Figure 4 genes-14-01956-f004:**
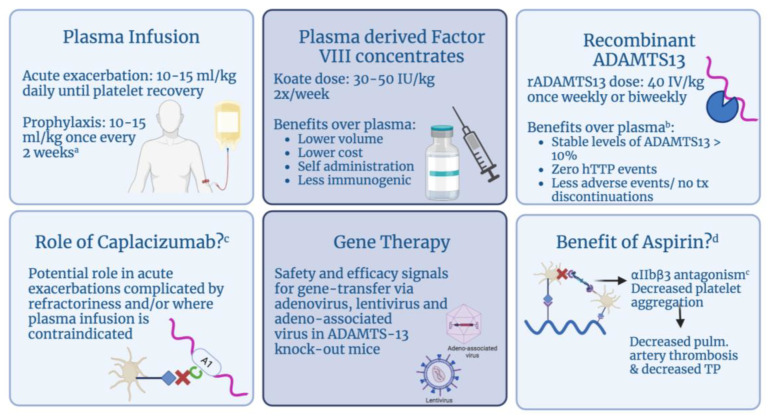
Therapeutic agents for hereditary thrombotic thrombocytopenic purpura (hTTP). ^a^ Prophylaxis in pregnancy is recommended to begin when pregnancy is confirmed, until 6 weeks postpartum. ^b^ Interim results from recent phase 3 clinical trial presented at ISTH 2023. ^c^ Of note, caplacizumab is not used as standard treatment for hTTP. Its utility has been described in an isolated case report and we question whether it can be used in rare cases where first-line therapies have failed and/or are contraindicated. ^d^ Aspirin prevents platelet aggregation by blocking the activation of platelet fibrinogen receptor αIIbβ3. These correlations (depicted in the figure) suggest that aspirin may be effective in preventing thrombosis in hTTP remission [41]. Abbreviations: rADAMTS13: recombinant ADAMTS13; pulm: pulmonary.

**Table 1 genes-14-01956-t001:** Comparison of key distinguishing clinical features of hereditary thrombotic thrombocytopenic purpura (hTTP) and ABO incompatibility [16,17]. Table reused from James George’s publication (*Res Pract Thromb Haemost*, 2022) [17].

Clinical Feature	Hereditary TTP (4 Infants)	ABO Incompatibility (20 Infants)
Family history	Two patients each had one older sibling who died 2 days after birth with jaundice, hemolysis	No infant deaths
Jaundice onset (h)	10 (5–13)	69 (18–82)
Bilirubin (mg/dl), maximum	24 (38 h after birth)	16 (74 h after birth)
Bilirubin response to phototherapy, IVIg ^a^	0	20 (100%)
Hemoglobin (g/dL, mean, minimum)	10.6	16.3
Platelet count (/μL, mean, minimum)	17,000	291,000

^a^ All patients with ABO incompatibility and hTTP were initially treated with phototherapy and IVIg.

## Data Availability

Not applicable.

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
