# Peer review of "Hereditary Thrombotic Thrombocytopenic Purpura"

_genes, 2023, doi:10.3390/genes14101956_

Round 1

Reviewer 1 Report

The article provides a very clear and complete explanation of the diagnostic and therapeutic progress related to HTTP, which is very valuable for clinical practice. Agree to publish.

Author Response

Thank you. 

Reviewer 2 Report

The article entitled “Hereditary thrombotic thrombocytopenic purpura” focuses on an rare hereditary disease with high thrombotic risk named hereditary thrombotic thrombocytopenic purpura (hTTP). This article is well-structured in terms of pathogenetic informations and diagnostic tests. In addition,  the authors report clinical cases that provide an useful guide in the diagnosis and treatment. The purpose of this review is satisfied because  it helps the clinician in the differential diagnosis with other similar thrombotic disease such as the autoimmune TTP and stimulates. I think that this article is suitable for publication in its current version.

Author Response

Thank you for your feedback.

Reviewer 3 Report

In their manuscript „Hereditary thrombotic thrombocytopenic purpura“ Nusrat et al summarize the complete literature that has been focused on this topic over the last years. The paper is very well written and I enjoyed reviewing this manuscript and can appreciate the amount of work and diligence it took to put this comprehensive dataset together. The topic is of clinical relevance and the results primarily appear to be of importance to further clinical understanding and treatment. I especially enjoyed reading the inserted clinical cases.

Still some of the following content needs improvement or at least clarification.

Captions need some minor layout editing as justify print.

Clinical case #4 needs revision as some stage direction has been imprinted (- is there ADAMTS13 level available for brother). Readability concerning this case can be improved. 

Minor spelling mistakes should be adressed for the final version. 

Author Response

Diagnosis and Therapies for Genetic Diseases, September 2023

RE: Nusrat, et al., Hereditary thrombotic thrombocytopenic purpura

   Thank you and the reviewers for the constructive comments about our manuscript. We have responded to each of the reviewer’s comments below and we have revised our manuscript to address the reviewer’s comments and suggestions.

   These revisions are described in our response to the reviewer’s questions and statements below.

Reviewer 3; Comments to the Author

In their manuscript “Hereditary thrombotic thrombocytopenic purpura“ Nusrat et al summarize the complete literature that has been focused on this topic over the last years. The paper is very well written and I enjoyed reviewing this manuscript and can appreciate the amount of work and diligence it took to put this comprehensive dataset together. The topic is of clinical relevance and the results primarily appear to be of importance to further clinical understanding and treatment. I especially enjoyed reading the inserted clinical cases.

Recommendations/comments:

  1. Captions need some minor layout editing as justify print. The captions have been edited.

  1. Clinical case #4 needs revision as some stage direction has been imprinted (- is there ADAMTS13 level available for brother). Readability concerning this case can be improved. Thank you for pointing out that error, we have now corrected it. ADAMTS13 level for the brother has now been mentioned in the case. We have also edited the case to read better.

Thank you, 

Sanober Nusrat, MD

Reviewer 4 Report

Very nice review with interesting cases.

1. Which method was used to assess the activity of ADAMTS-13?

2. Any difference in the clinical presentations of different mutations (e.g.  c.1378 mutation)?

3. Any recommendations on the diagnostic algorithm of hTTP in countries with higher frequency (e.g.; Norway)?

4. Any relation or correlation between ABO blood group and ADAMTS-13 activity?

4. What is the authors' opinion about caplacizumab in hTTP? I am not in favor of its use in hTTP.

Minor editing corrections are needed.

Author Response

Diagnosis and Therapies for Genetic Diseases, September 2023

RE: Nusrat, et al., Hereditary thrombotic thrombocytopenic purpura

   Thank you and the reviewers for the constructive comments about our manuscript. We have responded to each of the reviewer’s comments below and we have revised our manuscript to address the reviewer’s comments and suggestions.

   These revisions are described in our response to the reviewer’s questions and statements below.

Reviewer 4; Comments to the Author

  1. Which method was used to assess the activity of ADAMTS-13? Florescence Resonance Energy Transfer (FRET) assay is the most commonly available commercial method for quantification of the ADAMTS13 activity. The cases described in this review had testing completed using this assay. This information has been added to the manuscript.
  2. Any difference in the clinical presentations of different mutations (e.g.  c.1378 mutation)? Additional evidence has been cited in the Pathophysiology
  3. Any recommendations on the diagnostic algorithm of hTTP in countries with higher frequency (e.g.; Norway)? We will plan to incorporate this in our future publications on hTTP but consider it to be beyond the scope of this review.
  4. Any relation or correlation between ABO blood group and ADAMTS-13 activity? To our knowledge, no study has studied the relationship between ABO blood group and hTTP. This could be because of the inherent rarity of this condition. There is some data about blood group-O being protective in autoimmune TTP.
  5. What is the authors' opinion about caplacizumab in hTTP? I am not in favor of its use in hTTP. The authors of this review would recommend exploration of other strategies before using Caplacizumab for hTTP and agree that its role is limited as standard of care.

Thank you, 

Sanober Nusrat, MD
